# Sarcopenia Identification during Comprehensive Geriatric Assessment

**DOI:** 10.3390/ijerph19010032

**Published:** 2021-12-21

**Authors:** Krzysztof Pachołek, Małgorzata Sobieszczańska

**Affiliations:** Clinical Department of Geriatrics, Wroclaw Medical University, ul. M. Curie Skłodowskiej 66, 50-369 Wroclaw, Poland; wl-31@umed.wroc.pl

**Keywords:** epidemiology, geriatrics, sarcopenia, body composition, activities of daily living

## Abstract

Comprehensive geriatric assessment (CGA) is a multidimensional diagnostic process enabling evaluation of elderly patients’ physical and mental health status that implies implementation of the management targeted on the preservation of functional independence. Sarcopenia is a common but often underdiagnosed geriatric syndrome associated with increased likelihood of functional dependence and mortality risk. The main objectives of the study were the evaluation of sarcopenia prevalence in the patient group subjected to CGA with the upgraded EWGSOP2 algorithm considering muscle strength as the key criterion and usage of bioimpedance (BIA) muscle mass assessment. The study group consisted of 101 patients (76 women and 25 men) admitted for planned CGA to the Geriatrics Department of Wroclaw University Hospital. A diagnosis of sarcopenia was made according to the EWGSOP2 protocol. Body composition was determined with the bioimpedance technique. Functional status was assessed with ADLs from the VES-13 scale and additional questions. Sarcopenia was diagnosed in 16.8% of the study participants. Sarcopenic individuals presented worse functional status and impaired social activity. Muscle strength, gait speed and muscle mass below cut-off values were associated with dependence found in ADLs. Results showed that sarcopenia is a common impairment correlated with worse functional status and vulnerability to adverse outcomes. BIA can be treated as an accessible and accurate technique for muscle mass measurement in screening for sarcopenia, and the EWGSOP2 algorithm should be an essential part of the routine CGA procedure.

## 1. Introduction

Comprehensive geriatric assessment (CGA) is a multidimensional diagnostic process focused not only on the evaluation of elderly patients’ health condition but also their functional and psychosocial status. The objectives of CGA are not restricted only to the chronic diseases and geriatric syndromes but also encompass treatment interventions which are targeted on the improvement of patients’ quality of life and delaying occurrence of functional independence [1]. Such goals are achieved owing to the utility of specifically designed scales and tools that facilitate the process and objectify the results obtained [2].

One of the leading challenges of CGA is screening for syndromes particularly affecting the geriatric population, such as dementia, falls, urinary incontinence and frailty. CGA can be performed in various settings including outpatient facilities, hospital wards and emergency units, likewise during an acute admission or throughout a planned stay in special dedicated units called GEMU (geriatric evaluation and management unit) [3].

Cochrane meta-analysis including 29 trials with 13,766 participants has confirmed the efficacy of CGA in decreasing mortality and nursing home admissions of geriatric patients [4]. CGA requires the cooperation of various health care specialists with a geriatrician as a head and coordinator of the team. Such requirements make CGA a time-consuming and expensive procedure which results in low-quality evidence for its cost-effectiveness [4]. Thus, the proper choice of diagnostic tests in CGA programs is of a great importance. The evaluation should address the chief health issues affecting the quality of life of a diagnosed patient.

Sarcopenia as a geriatric syndrome was defined for the first time by Irwin Rosenberg in 1988. The term originates from the combination of two Greek words *sarx* and *penia*, meaning flesh and loss [5]. Since that time many working groups have dealt with sarcopenia worldwide. These scientific groups were focused on improving the knowledge on the pathophysiology and clinical relevance of sarcopenia as well as elaborating the effective diagnostic algorithms. In the results, varying definitions and diagnostic criteria were published. The lack of a unified, commonly accepted definition of sarcopenia hampers estimation of epidemiology and related health outcomes of this condition. 

Recently, the revised recommendations, including results of up-to-date clinical trials, were published in 2019 by the European Working Group on Sarcopenia in Older People (EWGSOP2). The document defines sarcopenia as “a progressive and generalized skeletal muscle disorder that is associated with increased likelihood of adverse outcomes including falls, fractures, physical disability and mortality”. The EWGSOP2 diagnostic algorithm focuses on low muscle strength as a key diagnostic criterion, and the low muscle quality/quantity (mass) is acknowledged as a confirmation criterion. Additional detection of low physical performance indicates severe sarcopenia [6].

From the above-mentioned criteria, the assessment of muscle mass seems to be the most cumbersome task. Traditionally, the highest precision is attributed to computed tomography (CT) and magnetic resonance imagining (MRI), but the cost of a single test makes their utility in a clinical practice limited. Dual X-ray absorptiometry is less expensive but also requires a radiologist’s participation, which makes the procedure time-consuming [7]. In recent years, much attention was paid to a bioimpedance analysis (BIA), which offers non-invasive and quick estimation of muscle quantity pronounced as appendicular skeletal muscle mass (ASM) adjusted for height (ASM/h2) [8]; furthermore, this method also measures other body composition parameters, such as lean body mass (LBM), fat mass and phase angle (PA). 

Considering the fact that sarcopenia is an independent risk factor of falls, disability and even death [9,10] in a geriatric population, early diagnosis is vital. Regular physical activity, including progressive strength exercise, and a proper nutrition plan with a high protein intake play a crucial role in rebuilding muscle mass and strength [6]. Even patients in the tenth decade of life can receive a benefit from such interventions [11].

Data on sarcopenia epidemiology varies depending on the algorithms and thresholds used in different studies [12]. In the European population aged 65–94, sarcopenia is diagnosed in 10.2% of community-dwelling individuals [13], reaching 29% in the long-term care facilities residents [14]. Population aging will result in 72.4% growth in sarcopenia prevalence in European Union countries by 2045 [15].

The primary aim of the study was the estimation of sarcopenia prevalence in CGA patients with the latest EWGSOP2 algorithm. The study was also designed to assess the BIA utility as a cheap and quick method for muscle mass assessment. Additionally, it was planned to estimate the role of low PA values as indicators of sarcopenia and their correlation with patients’ functional status and vulnerability.

## 2. Materials and Methods

Patients were tested for sarcopenia with the EWGSOP2 algorithm [6]. Grip strength was measured with a JAMAR dynamometer according to the Southampton standardized approach [16]. Confirmation of sarcopenia diagnosis was made by estimation of ASM/h2 with a Tanita MC-980 body analyzer using bioimpedance analysis (BIA). BIA is based on measurement of body impedance which is the derivative of resistance (R) and reactance (Xc). The device uses alternating electrical current (AC) which changes its PA and voltage during the movement of charge through the patient’s body. Fat tissue and extracellular fluids have their own R, while cell membranes act as capacitors which means they have their own Xc. R causes decline in voltage and Xc decreases PA. Total body impedance varies depending on sex, race, and age. To make calculations possible, the patient’s body is treated as 5 cylinders (4 limbs and the trunk) connected in series. Further use of mathematical formulas based on data from anthropometric studies makes it possible to estimate the content of water and fat in the body and consequently muscle mass [17,18]. In assessment of sarcopenia severity, gait speed was used as an indicator of low physical performance. It was calculated on the distance of 6 meters and the average value from two measurements was considered.

The patient’s body height was measured in standing position with a stadiometer. Threshold values came from EWGSOP2. The sarcopenia diagnostic protocol used in the study is shown in Figure 1.

In the assessment of functional status, ADLs from the vulnerable elderly scale (VES-13) [19] with additional questions about history of falls, social activity and independent meal preparation were used.

The total score in the VES-13 scale was used in predicting the risk of adverse outcomes (vulnerability), as this questionnaire is a well-established predictor of death and functional independence in community-dwelling elderly patients [20,21].

The study group was recruited from patients hospitalized in the Geriatrics Department of Wroclaw University Hospital. All participants had scheduled admission for CGA with a referral from the family doctor, were above 65 years old and without acute health disorders. Patients with severe dementia, immobility, and implantation of electronic devices (the risk of interference with BIA electrical field) were excluded from the sarcopenia evaluation. 

A statistical analysis was performed using Microsoft Excel and Tibco Software Statistica version 13.3 (TIBCO Software Inc., Palo Alto, CA, USA). The distribution of values was verified by the Shapiro–Wilk test. Age values following normal distribution were compared with Student’s *t*-test. All other values, as randomly selected, were compared with Mann–Whitney’s U test. Rank correlation of values was verified with Spearman’s rank correlation coefficient (R). Verification of association between categorical values was carried out with the Chi-square test of independence. A *p* value < 0.05 was considered statistically significant in all analyses.

The study protocol was approved by the Bioethics Committee of Wroclaw Medical University (No. KB-513/2018). All participants signed the written and prior informed consent following the international norms and procedures of medical research in humans using the Helsinki Declaration.

## 3. Results

The examined group consisted of 101 patients, of which 76 were women and 25 were men. The study group characteristics are presented in Table 1.

Sarcopenia was diagnosed in 17 patients (16.8%), among whom 6 had severe sarcopenia. Prevalence of sarcopenia was higher in the examined men, as compared to women (Table 2).

The mean age of sarcopenic patients was slightly higher than in non-sarcopenic individuals (78.5 vs. 81), but the difference was not statistically significant. Sarcopenic patients reached significantly higher results in VES-13, independent of the age points. There was no significant difference in terms of BMI and PA between the two groups (Table 3).

The patients with sarcopenia presented worse functional status. Of the ADLs assessed in the study, they had more difficulties with reaching out above the shoulder, walking a distance of 400 m and performing hard household tasks. They also needed assistance more often in light housework and in using the bathroom or showering. They were also less likely to be able to prepare meals by themselves. Moreover, a diagnosis of sarcopenia was associated with poorer social activity pronounced by lower frequency of visiting family members and friends (Table 4).

Patients with a grip strength below the EWGSOP2 cut-off values had a significantly higher score in VES-13. Out of the BIA body composition parameters, only low muscle mass correlated with low muscle strength. There was no difference in age and BMI between patients with low muscle strength and normal muscle strength (Table 5).

There was a positive correlation between muscle strength and mass, lean body mass (LBM), %LBM, %muscle mass, ASM/h^2^, PA and gait speed. There was a negative correlation of muscle strength with VES-13 score and %fat mass (Table 6).

Patients with low gait speed (according to EWGSOP thresholds) had a significantly higher score in VES-13. There was no difference in age, BMI, and body composition parameters between patients with low gait speed and those with normal gait speed. Contrary to this, there was a positive correlation between gait speed and grip strength, and a negative correlation with VES-13 score (Table 7).

The ASM/h^2^ values below the EWGSOP2 cut-off values was associated with higher VES-13 scores and lower BMI values. There was a positive correlation between ASM/h^2^ and mass, fat mass, BMI, PA, and grip strength (Table 8).

## 4. Discussion

A comparison of sarcopenia epidemiology between the various studies is hampered by varying diagnostic criteria and different properties of the study groups analyzed. Moreover, at the moment of preparing this publication, the authors did not find any other studies assessing the epidemiology of sarcopenia in Polish patients of geriatric or internal medicine departments. The study is also unique in establishing sarcopenia prevalence using the EWGSOP2 algorithm during the CGA procedure. Data from the study concerning the group of Polish community-dwelling individuals aged above 65 showed a presence of sarcopenia in 19.4% of them [22]. Interestingly, in the study, where muscle mass was estimated from an equation based on BMI, sarcopenia was diagnosed only in 12.6% of Polish elderly individuals [23] against 16.8% in the present study. Therefore, in our opinion, a measurement of muscle mass can increase the frequency of sarcopenia detection. In the study on 198 German patients in geriatric departments, sarcopenia was diagnosed in 25.3% of them [24]. The higher prevalence of sarcopenia observed in that study might have been a result of the emergency hospital admission, which may be an indicator of worse general health condition of participants, in contrast to our community-dwelling patients who were assessed during the planned diagnostic hospitalization.

Sarcopenia is a well-known risk factor of functional status deterioration in the elderly. For the assessment of independence in ADL, the VES-13 questionnaire was applied. This tool covers patients’ age, self-rated health status, as well as basic and instrumental (associated with running household) ADLs. The correlation of the worse result in VES-13 (higher scores) with sarcopenia was independent of the scoring obtained by the examined patients because of age. The selection of ADLs evaluated in VES-13 is well-suited for community-dwelling elderly people, which is confirmed by studies showing a correlation of higher VES-13 with mortality and disability in this population [21]. Using scales more focused on basic ADL, such as Barthel’s scale, may result in omitting the influence of sarcopenia on the patients’ functional status [24]. In turn, other studies which evaluated more advanced ADLs support the findings of our study [25]. It is worth highlighting that all the sarcopenic patients in the present study had at least three points in VES-13. This fact is considered related to vulnerability for adverse outcomes [19]. In addition, sarcopenic patients had more difficulties with performing nearly all the ADLs evaluated in VES-13 (except money management). The median VES-13 score was also higher in a group with low grip strength compared to groups with slow walking speed and low ASM/h2. This observation was also detected in prospective studies [26,27] and endorses the rationale behind modifications in the updated EWGSOP2 diagnostic algorithm, which made muscle strength the key criterion.

The percentage of patients with sarcopenia declaring their health status as poor was similar in the non-sarcopenic group (35% vs. 32.6%). This indicates a necessity of active sarcopenia screening in geriatric patients, even in patients who do not report complaints about their functional abilities. 

Poorer social activity of sarcopenic patients was observed also in other cross-sectional studies and may be related to impaired mobility and fear of falling [28]. It is worth emphasizing that the lack of participation in social life leads to anxiety and depression significantly affecting the quality of life in seniors [29].

Muscle mass was estimated with BIA, the most prominent advantage of which is simplicity of testing procedure and immediate obtainment of the results. Owing to the instant delivery of ASM/h2 by BIA analyzer, it took approximately 15 minutes in each patient to confirm sarcopenia diagnosis and then assess its severity. Such a quick testing method is especially applicable in procedures such as CGA, when in the relatively short time many screening tests shall be provided. Additionally, BIA devices are relatively cheap and portable. The accuracy of BIA was confirmed in comparative studies with DXA [30] and MRI [31] considered to be benchmark methods. DXA, MRI and CT seem to be too costly and time-consuming to be used on a large scale in everyday clinical practice. Nonetheless, BIA has its limitations. Firstly, BIA results are not based on direct measurement of body composition but on its estimation with mathematical formulas and data from anthropometric studies [32]. These formulas may differ depending on the device producer; thus, the accuracy of particular models can be uncertain in specific tested populations. The BIA results can be influenced by many factors. Recent food intake may result in overestimation of total body water and consequently ASM/h2. The same mechanism is present in patients with fluid overload. Underestimation of ASM/h2 can be observed after intensive exercise or in dehydration. Other factors affecting the precision are erroneous height measurement or serious limb deformations altering the patient’s posture [33]. For that reason, all BIA tests in this study were performed at least 4 hours after the last meal. The equations used by the device were validated within the European elderly population. 

The patients with low ASM/h2 had significantly higher VES-13 scores, which supports the evidence for association of low ASM/h2 with various adverse outcomes, such as death or disability [34,35] and confirms the clinical significance of cut-off values recommended by the EWGSOP2. 

These results confirm that BIA is a cheap and time-efficient method in confirming the diagnosis of sarcopenia, thus, the study may contribute to increased detection of sarcopenia in clinical practice. 

PA is calculated by all BIA devices; this is a mathematical function of reactance illustrating a phase shift in alternating electrical current. PA is constant for every patient at the given moment, and it should be reproducible independent of the BIA device model. High PA values reflect the number of cells in the organisms, because cellular membranes act as an electric capacitor increasing PA. PA usually decreases with age or during inflammatory diseases and malnutrition. Interestingly, the highest PA values were observed in professional athletes [36]. Prospective trials have confirmed predicting the PA value for mortality in hospitalized geriatric patients [37,38]. 

This study detected a positive correlation of PA and ASM/h2, which was also observed in other studies [39,40]. There was no significant difference between sarcopenic and non-sarcopenic patients. In addition, there was no correlation of PA with gait speed and muscle strength. According to these results, PA does not seem to be a useful parameter in sarcopenia screening. However, a negative correlation of PA with VES-13 score can be applicable in the detection of vulnerable patients during CGA. 

The most important limitation of this study is its cross-sectional character which precludes approving a cause–effect relationship between the reported correlations. Data on functional status was self-reported, thus susceptible for a possible simulation or dissimulation.

## 5. Conclusions

Sarcopenia is a common impairment in elderly patients that is associated with a worse functional status and may be considered as a predictor of adverse medical outcomes in this population. Thus, evaluation of sarcopenia should be recommended as an obligatory component of CGA. The study results show the necessity of active sarcopenia searching during the CGA procedure. The EWGSOP2 diagnostic algorithm with handgrip strength measurement and assessment of muscle mass with BIA appears to be a feasible tool for this purpose. BIA provides an accessible and accurate technique for muscle mass measurement in confirming a sarcopenia diagnosis. PA can be used as an additional parameter in estimating elderly patients’ vulnerability.

## Figures and Tables

**Figure 1 ijerph-19-00032-f001:**
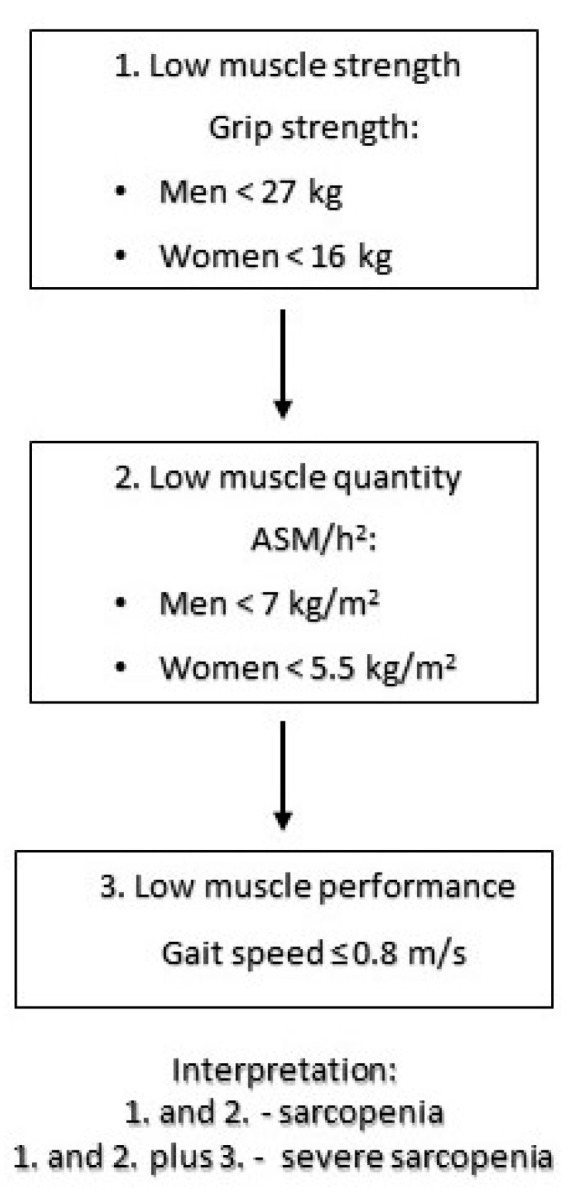
Sarcopenia diagnostic algorithm used in the study based on EWGSOP 2 recommendations [6].

**Table 1 ijerph-19-00032-t001:** Characteristics of study group.

	Overall (n = 101)	Women (n = 76)	Men (n = 25)
Average Age	78.53	78.79	77.76
Median Age ± SD	79 ± 6.79	79 ± 6.62	79 ± 7.2

**Table 2 ijerph-19-00032-t002:** Prevalence of sarcopenia and its diagnostic criteria.

	Overall	Women	Men
Sarcopenia	17 (16.8%)	10 (13.2%)	7 (28%)
Severe sarcopenia	6 (5.9%)	4 (5.3%)	2 (8%)

**Table 3 ijerph-19-00032-t003:** Comparison of sarcopenic and non-sarcopenic patients.

	Sarcopenic (n = 17)			Non-Sarcopenic (n = 84)			*p*
	Mean	Min.	Max.	Mean	Min.	Max.	
VES-13	6.47	3.00	9.00	3.75	0.00	10.00	<0.001
BMI (kg/m^2^)	24.84	18.33	37.96	27.53	15.98	47.29	0.18
Fat mass (%)	27.38	8.10	67.49	32.66	8.70	51.22	0.03
Phase angle (φ)	4.41	3.00	8.00	4.55	3.60	6.80	0.5
Age	81.18	71.00	94.00	78.00	65.00	90.00	0.14

**Table 4 ijerph-19-00032-t004:** Comparison of functional status between sarcopenic and non-sarcopenic patients.

	Sarcopenic Patients	Non-Sarcopenic Patients	*p*
VES-13 Activities			
Difficulty with			
Kneeling, bending and stooping	12 (70.6%)	44 (52.4%)	0.62
Lifting and carrying 5 kg	9 (52.9%)	25 (29.8%)	0.07
Reaching out and lifting upper extremities above the shoulder	9 (52.9%)	21 (25%)	0.02
Writing or handling and grasping small objects	5 (29.4%)	13 (15.5%)	0.17
Walking the distance of 400 m	10 (58.8%)	22 (26.2%)	0.01
Performing hard housework	17 (100%)	34 (40.5%)	<0.001
Needs assitances with			
Shopping	5 (29.4%)	12 (14.3%)	0.13
Money management	0	1 (1.19%)	-
Transfer	0	0	-
Light housework	5 (29.4%)	6 (7.1%)	0.01
Bathing and showering	8 (47.1%)	6 (7.1%)	<0.001
Additional questions			
History of falls within last 12 months	11 (64.7%)	38 (45.2%)	0.14
Avoidance of visiting friends or family members	9 (52.9%)	21 (25%)	0.02
Dependance in meal preparation	6 (35.3%)	2 (2.4%)	<0.001

**Table 5 ijerph-19-00032-t005:** Comparison of patients with low grip strength to patients with normal grip strength.

	Low Grip Strength (n = 19)			Normal Grip Strength (n = 82)			*p*
	Mean	Min.	Max.	Mean	Min.	Max.	
VES-13	6.65	3.00	9.00	3.81	0.00	10.00	<0.001
BMI (kg/m^2^)	26.80	18.33	37.96	27.12	15.98	47.29	0.6
Muscle mass (%)	61.20	56.10	87.30	69.20	56.11	87.27	0.03
Fat mass (%)	28.86	8.10	67.49	32.27	10.10	51.22	0.09
ASM/h^2^ (kg/m^2^)	7.22	5.98	10.73	7.65	5.98	10.07	0.55
Phase angle (φ)	4.74	3.00	8.00	4.49	3.60	6.80	0.92

**Table 6 ijerph-19-00032-t006:** Correlation of sarcopenia diagnostic criteria.

	Grip Strength		Gait Speed		ASM/h^2^	
	Spearman Coeff.	*p*	Spearman Coeff.	*p*	Spearman Coeff.	*p*
Gait speed (m/s)	0.219	0.028	-	-	0.447	<0.001
VES-13	−0.521	0.000	−0.559	0.000	0.301	0.01
Body mass (kg)	0.335	0.001	0.023	0.817	−0.165	0.1
BMI (kg/m^2^)	0.121	0.226	−0.054	0.589	0.705	<0.001
Fat mass (%)	−0.284	0.004	0.079	0.434	0.689	<0.001
LBM (kg)	0.557	0.000	−0.044	0.665	0.062	0.54
LBM (%)	0.216	0.030	−0.134	0.182	0.756	<0.001
Muscle mass (%)	0.221	0.027	−0.134	0.180	−0.070	0.49
ASM/h^2^ (kg/m^2^)	0.353	0.000	−0.162	0.105	-	-
Phase angle (φ)	0.245	0.014	−0.026	0.800	0.317	<0.001

**Table 7 ijerph-19-00032-t007:** Comparison of patients with low gait speed to patients with normal gait speed.

	Low Gait Speed (n = 27)			Normal Gait Speed (n = 74)			*p*
	Mean	Min.	Max.	Mean	Min.	Max.	
VES-13	5.57	1.00	9.00	3.02	0.00	10.00	<0.001
BMI (kg/m^2^)	27.27	18.33	38.33	26.91	15.98	47.29	0.75
Muscle mass (%)	64.29	53.90	87.27	60.74	46.30	85.15	0.31
Fat mass (%)	28.86	8.10	67.49	32.27	10.10	51.22	0.12
ASM/h^2^ (kg/m^2^)	7.52	5.61	10.58	7.08	5.69	10.73	0.16
Phase angle (φ)	4.61	3.00	8.00	4.45	3.60	6.80	0.67

**Table 8 ijerph-19-00032-t008:** Comparison of patients with to patients with normal ASM/h2.

	Low ASM/h^2^ (n = 19)			Normal ASM/h^2^ (n = 82)			*p*
	Mean	Min.	Max.	Mean	Min.	Max.	
VES-13	5.89	2.00	8.00	4.04	0.00	10.00	0.03
BMI (kg/m^2^)	19.89	18.33	28.82	27.78	15.98	47.29	<0.001

## Data Availability

The data that support the findings of this study are available on request from the corresponding author, K.P.

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
