# Peer review of "Sarcopenia Identification during Comprehensive Geriatric Assessment"

_ijerph, 2021, doi:10.3390/ijerph19010032_

Round 1
Reviewer 1 Report
The present study was diagnosed the sarcopenia status in the CGA patients. The study was well conceptualized and carried out. And the conclusion is appropriate from the authors perspective. However, the authors need to improve in some aspect as given below before the publication.
1. For example, several statements in the introduction and discussion do not have proper reference citations. Add the reference for supporting the statement (line no91-92), Add the reference for the following text “The comparison of sarcopenia epidemiology between other studies is hampered by 174 various diagnostic criteria and different properties of study groups.” These must be given with proper references.
2. Authors mentioned that greater prevalence of sarcopenia in this study may be a result of higher median age which is contrast to their own study. What the authors were intent to mean? What it Contrast refere here?
3. In the table 4, light house work (eg. Dusting), which means cleaning or what ?
Author Response
We are grateful for your review of our study and appreciate that you took time to evaluate our manuscript. We have addressed all the concerns raised in the review.
- The reference supporting the statement in lines 91 and 92 was added. Dodds, Richard Matthew, et al. "The epidemiology of sarcopenia." Journal of Clinical Densitometry4 (2015): 461-466.
- In part of the discussion explaining differences in sarcopenia incidence between this and similar study (https://doi.org/10.1016/j.jamda.2013.11.027) , the matter of median age was removed as discrepancies in median age in both study groups are not statistically significant. The issue of type of hospital admission was emphasized as most probable reason for higher incidence of sarcopenia in the aforementioned study.
- The examples of hard and light house works were removed from table 4., to preserve the literal form of questions used in English version of VES-13 form.
Kind regards,
Krzysztof Pachołek
Reviewer 2 Report
Thank you for the opportunity to review the manuscript entitled Sarcopenia Identification During Comprehensive Geriatric Assessment.
GENERAL COMMENTS:
- The overall objective of this study was “evaluation of sarcopenia...". In this way, the evidence shows that decreased muscle strength has been better associated with adverse clinical outcomes; therefore, muscle strength is the first criterion in the sarcopenia algorithm for case-finding. However, the authors emphasize in the introduction, discussion, and conclusion, mainly the evaluation of muscle mass by BIA, leaving muscle strength in the background.
- All tables of results are presented in a format that is difficult to visualize and understand. I suggest changing to a more appropriate layout: place the grouping variables in the columns and the variables to be compared in the rows.
- The means and medians are very similar in all result tables, which may indicate a normal distribution. Why are all results reported with nonparametric Mann-Whitney U test?
Line 7: If you use abbreviations in the abstract, try to define them when writing them for the first time. Remember that often only the abstract is accessible and must be understandable.
Lines 10-16: if the study purpose was the assessment of sarcopenia. Why do the authors only give details of the evaluation of muscle mass by BIA and do not do the same with the assessment of muscle strength?
Line 11: A study design like this, could assess the incidence of sarcopenia? Maybe the authors mean prevalence?
Lines 52-63: This entire paragraph should be referenced by "6" and not just half of it. (I suggest putting the reference at the end of the paragraph).
Line 70: The use of terms should be according to the proposed references. Why do the authors decide to use the term "skeletal muscle index SMI" if both the cited reference "8" and the cut-off points (EWGSOP2) used: "appendicular skeletal muscle mass (ASMM) or appendicular skeletal muscle mass index"?
Lines 73-75: It is not clear what the authors want to express. It should be understood that the definition of sarcopenia has evolved from a geriatric syndrome to a muscular disorder.
Lines 87-89: Regarding sarcopenia and loss of muscle mass; “the effect of progressive strength exercise on muscle mass” is the correct statement and not just regular physical activity.
Line 110: Provide a reference to the algorithm used for sarcopenia diagnosis. If it is from EWGSOP2, I suggest changing SMI to ASM/h2.
Lines 205-214: This paragraph corresponds to the methodology section.
Line 260: Primary, the authors, should propose including the assessment of muscle strength (handgrip) in evaluating sarcopenia. The BIA is a diagnostic tool to confirm sarcopenia, and its use is not as screening, (EWGSOP2).
Author Response
We are grateful for your review of our study and appreciate that you took time to evaluate our manuscript. We have addressed all the concerns raised in the review.
- The primary role of muscle strength assessment in diagnosing sarcopenia is well-established by many cross-sectional and observational studies, which has been expressed by the recent EWGSOP2 algorithm. Confirmation of sarcopenia diagnosis, according to the EWGSOP2, mandates evaluation of muscle mass. “Golden standard” techniques enabling assessment of muscle mass like MRI, CT or DXA are costly and time-consuming. This is why we have decided to choose BIA which allows for relatively cheap and quick confirmation of sarcopenia diagnosis, enabling assessment of sarcopenia in large groups of patients in a short period of time. The introduction and discussion sections along with conclusion paragraph have been modified to pronounce it more clearly.
2. All the tables in the manuscript were modified to clearer layout. Grouping variables were put in the columns as suggested. - Distribution of all the values was checked with Shapiro-Wilk test. Age of patients was the only value following the normal distribution, therefore the age of sarcopenic to non-sarcopenic group was compared with the t-Student’s test. There was a mistake in the table suggesting that the Mann-Whitney test was used. Paragraph related to statistical analysis in the materials and methods section was modified to clarify this issue.
Line 7: The CGA shortcut is now defined.
Lines 10-16: The text was modified to emphasize that the evaluation of sarcopenia was performed with the EWGSOP 2 algorithm, which uses muscle strength as key criterion and muscle mass as confirmation of sarcopenia diagnosis.
Line 11: This is cross-sectional study, thus it is not able to assess sarcopenia incidence. The objectives have been corrected.
Lines 52-56: The reference has been put at the end of the paragraph.
Line 70: The term SMI (skeletal muscle index) was replaced by ASM/h2 (appendicular skeletal muscle mass index) to make the terminology compliant with the EWGSOP2 recommendations.
Lines 73-75: The whole paragraph was removed from the introduction since pathomechanism of sarcopenia is beyond the scope of this study.
Lines 87-89: The paragraph considering sarcopenia prevention and treatment has been corrected according to the Reviewer’s suggestion.
Line 110: The reference to the EWGSOP2 recommendations has been added to the description of Figure 1.
Lines 205-214: The paragraph about BIA has been moved to the materials and methods section.
Line 260: The paragraph with conclusions of the study was corrected to emphasize the results of the study more accurately.
Kind regards,
Krzysztof Pachołek
Reviewer 3 Report
This paper addresses sarcopenia, which is a major area of concern in today's society. The authors evaluate the incidence of sarcopenia in patients undergoing a comprehensive geriatric assessment, including an algorithm and muscles mass scores. They also attempt to identify individuals who may be at higher risk. However, it is unclear what the direct relevance of these results is. What new information do they contribute to the field? Much of the data showing worse functional status and impaired activities is not novel for individuals with sarcopenia. The authors should address the comments below, considering how this particular study significantly advances the study of sarcopenia.
Major comments:
- Be sure to define all abbreviations at the first use, including in the abstract.
- There seem to be a large number of "the" articles missing. There are also numerous other awkward expressions. Please have the manuscript language carefully checked by a native English speaker.
- Introduction: The introduction is overly extensive. Limit the information in the introduction to the material that directly supports the purposes of the study. The introduction should be cut by at least a third.
- Lines 96-100: The aims of the study are unclear. It seems that the aims are simply to gather information. The last aim (lines 99-100) seems to be more clear. However, this last aim needs more information.
- Paragraph starting on line 120: It is unclear how this statistical analysis tests the third aim. Please provide more information regarding the assessment of sarcopenia to detect vulnerable patients.
- Line 122: This should be "Student's t-test". In addition, what values were compared using this test?
- It is unclear how this study provides new information or advances the field, other than it was conducted in Poland (lines 177-178). The novelty and contribution of the study need to be explicitly explained.
- The discussion should focus more on the relationship of the current results to the existing literature. It focuses too much on providing a summary of the field rather than the contribution and relevance of the current results.
Minor comments:
- Lines 32 and 257: "which" should be "that". See https://www.grammarly.com/blog/which-vs-that/?gclid=CjwKCAiAs92MBhAXEiwAXTi253ZjeRCRdFjtKkPM4OAPvxPzWHeegRiH6pABCTnDzNvyDjYkavMuhRoCgrEQAvD_BwE&gclsrc=aw.ds.
- Line 56: The term "lately" does not make sense here. This should be "Recently".
Author Response
We are grateful for your review of our study and appreciate that you took time to evaluate our manuscript. We have addressed all the concerns raised in the review.
- The CGA shortcut used in the abstract has been defined. All the other shortcuts have been checked.
- The English language in the article has been thoroughly checked and corrections have been made.
- The introduction has been significantly shortened. The paragraph referring to sarcopenia pathomechanism along with other unsignificant expressions have been removed from the introduction as they are not directly related to the aims of the study.
- The paragraph concerning aims of the study has been modified to express them more precisely and clearly.
- The new paragraph regarding the vulnerable patients has been added (lines 133-135).
- Age values were compared with the Student’s t-test as their normal distribution was confirmed by Shapiro-Wilk test. The paragraph considering statistical analysis has been modified to inform which values were compared with the Student’s t-test.
- and 8. The discussion has been rewritten to express the novelty and contribution of the study more vividly. Paragraph explaining BIA method has been moved to the materials and methods section. Additional references comparing obtained results to other studies have been included.
Language errors highlighted by the Reviewer have been corrected along with other grammar mistakes.
Kind regards,
Krzysztof Pachołek
Round 2
Reviewer 2 Report
Thanks for the answers. The authors have done a good job revising the document. The tables are clear now. I have only a few (minor) comments to make:
- Lines 91-104 The authors use the EWGSOP2 criteria (and not the Asian criteria) to diagnose sarcopenia. Therefore, strength and muscle mass should place first. Physical performance is a criterion for sarcopenia severity. Therefore, it should be explained as a severity criterion at the end of the paragraph.
- Lines 241-242 (…) the assessment of all the three diagnostic criteria of sarcopenia (…)
In the same mind as the previous suggestion: according to EWGSOP2, there are two diagnostic criteria for sarcopenia. Physical performance is a criterion of severity in an already diagnosed sarcopenia. I understand that this difference, as shown in Table 2, was taken into account in all results. Therefore, I suggest the authors rewrite the cited text to avoid misinterpretation:
- Lines 288-290 (…) The EWGSOP2 diagnostic algorithm with handgrip strength measurement as screening test followed by BIA technique for assessment of body composition appears to be a feasible tool for this purpose. (…)
It can be inaccurate to address the handgrip as a screening tool (remembering that the SARC-F has this purpose). I suggest changing it as follows:
(…) The EWGSOP2 diagnostic algorithm with handgrip strength measurement and assessment of muscle mass by BIA appears to be a feasible tool for this purpose (…)
- It is always appropriate (and appreciated) that authors specifically cite the modified text in their responses.
Author Response
We are grateful for all your valuable and insightful comments that led to improvements in the current version of the manuscript.
Below we provide point by point responses to the Reviewer’s comments.
Lines: 91-104: Assessment of physical performance has been put at the end of the paragraph in the materials and methods section (after muscle strength and muscle mass measurement). The text has been modified to express the role of muscle performance criterion in the EWGSOP2 algorithm accurately.
“In assessment of sarcopenia severity gait speed as an indicator of low physical performance was used. It was calculated on the distance of 6 meters, average value from two measurements was taken into account.”
Lines 241-242: The text has been rewritten, according to the Reviewer’s suggestion, to avoid misinterpretation of the EWGSOP 2 algorithm,.
“Thanks to the instant delivery of ASM/h2 by BIA analyzer, it took approximately 15 minutes in each patient to confirm sarcopenia diagnosis and then assess its severity.”
Lines: 288-290:
The text of the paragraph has been modified to avoid misinterpretation that hand grip strength measurement is a screening test for sarcopenia.
“The EWGSOP2 diagnostic algorithm with handgrip strength measurement and assessment of muscle mass with BIA appears to be a feasible tool for this purpose.”
Kind regards,
Krzysztof Pacholek